# RaiseWikibase: Fast inserts into the BERD instance

Renat Shigapov[1]✉[0000−0002−0331−2558], Jörg Mechnich[1][0000−0002−6406−4906], and Irene Schumm[1][0000−0002−0167−3683]

Mannheim University Library, University of Mannheim, Germany
{firstname}.{lastname}@bib.uni-mannheim.de

**Abstract.** We create a knowledge graph of German companies in order to facilitate research with Business, Economic and Related Data (BERD), both modern and historical. For the implementation we chose Wikibase, but the wrappers of the Wikibase API turned out to be slow for filling it with millions of entities. This work presents the open source tool RaiseWikibase for speeding up data filling and knowledge graph construction by inserting data directly into the database. We test its performance for creating the items and wikitexts and share a reusable example for knowledge graph construction.

**Keywords:** Knowledge graph construction · Wikidata · Wikibase

## 1 Introduction

*Motivation.* German company data are spread over many providers, registers and time spans. The company identifiers in Germany are sadly famous for their lack of uniqueness, inconsistent representations and multiple registrations per legal entity[1]. The modern data for millions of German companies were scraped and unchained by OpenCorporates [9]. The collection was supported by the TheyBuyForYou project [8] and is used in euBusinessGraph [6]. Historical information about German companies is still confined within many undigitized documents as reported by the EurHisFirm project [5]. Only some of the documents were digitized, processed and structured [2,3]. All this makes knowledge graph construction for German companies difficult, urgent and necessary.

*Wikibase and Wikidata.* We chose Wikibase for creating a knowledge graph. Its benefits are a live Blazegraph-based SPARQL endpoint, RDF export, an API, data science tools and the promising strategy for the decentralized Wikibase Ecosystem [4]. The general-purpose Wikidata knowledge graph [10] (the main Wikibase instance) can help with ontology development for new instances.

---

[1] https://blog.opencorporates.com/2019/02/19/wait-what-the-problems-of-company-numbers-in-germany-and-how-were-handling-them

*Related work.* An ontology and data can be filled into a Wikibase instance manually or using the wrappers of the Wikibase API. WikidataIntegrator [1], wikibase-cli, Wikidata-Toolkit, WikibaseIntegrator, Pywikibot, QuickStatements and many other tools are excellent for data filling during the collaborative knowledge graph development. However, they can insert roughly 1-6 entities per second, making data filling and knowledge graph construction with a fresh Wikibase instance lengthy. A solution is to insert data directly into the database, but a ready-to-use tool for it does not yet exist. The only relevant work [11] provides code in Java for Wikibase 1.34. Unfortunately, its reuse requires changing hard-coded values and restructuring of the code. Instead, we implemented RaiseWikibase in Python for Wikibase 1.35.

*Our contribution and structure.* We present the tool RaiseWikibase for speeding up data filling and knowledge graph construction with Wikibase. Next, we describe RaiseWikibase, raise our BERD instance, and make conclusions.

## 2   Raising Wikibase

RaiseWikibase is written in Python, uses the version "1.35" of the Wikibase Docker image, and connects to the MariaDB database using the mysqlclient library. The open source code is shared at https://github.com/UB-Mannheim/RaiseWikibase.

The main functions are `page` and `batch`. The `page` function executes inserts into the database but does not commit them. Multiple `page` functions are wrapped into a transaction inside the `batch` function. Creating a million of items is as simple as `batch('wikibase-item',items)`, where the first argument specifies a content model and `items` is a list of JSON representations of the items.

To create or edit the JSON representation of an entity, we added the following functions for the Wikibase data model[2]: `entity`, `claim`, `snak`, `label`, `alias` and `description`. They return the suitable template dictionaries. Eighteen datatypes are implemented in the `snak` function. The JSON representation of an item with an English label, aliases, description and one claim is given by:

```
item = entity(labels=label('en', 'organization'),
              aliases=alias('en', ['organisation', 'org']),
              descriptions=description('en','social entity'),
              claims=claim(prop='P2101',
                           mainsnak=snak(datatype='string',
                                         value='org',
                                         prop='P2101')),
              etype='item')
```

RaiseWikibase inserts data into the nine tables according to the database schemas of Mediawiki and Wikibase. While this is sufficient in case of unstructured data, for structured data some of the secondary tables are also changed.

---

[2] https://www.mediawiki.org/wiki/Wikibase/DataModel

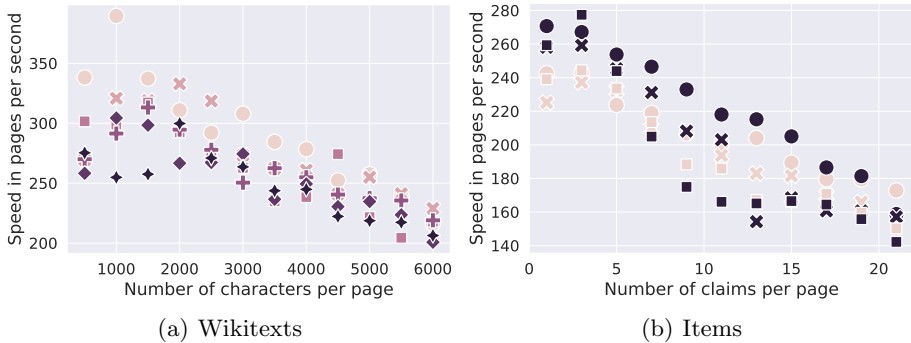

(a) Wikitexts  (b) Items

Fig. 1: RaiseWikibase performance in the batch mode of page creation.

Fig. 1 shows the results of our performance analysis. Every data point is based on a batch of ten thousands pages. Fig. 1a illustrates the results of six repeated experiments indicated by differently-shaped data points. In Fig. 1b two colors represent repeated experiments and three shapes of a data point stand for: ● – a claim lacks qualifiers and references, ✖ – a claim has one qualifier and no reference, and ■ – a claim has one qualifier and one reference. The insert rate decreases approximately linearly with increasing number of characters per wikitext and with increasing number of claims per item. Small pages can be uploaded at rates of 250-350 wikitexts per second (Fig. 1a) and 220-280 items per second (Fig. 1b). The source codes and technical details about our tests can be found on GitHub.

The ElasticSearch index and some of the secondary tables are built after data filling. Queries on the "page" and "text" tables can be made. A bot account, needed for the wrappers of the Wikibase API, is created automatically.

## 3 Raising BERD

A fresh Wikibase instance contains only the main page. An ontology, templates, modules, structured and unstructured data have to be filled into the database. Some extensions have to be installed and many parameters have to be configured.

We prepared the configuration files, modules and extensions, created the templates, changed the sidebar, a style of pages (skin) and the main page. The page with SPARQL examples, shown at the query frontend, is also created. These files are stored in the "texts" folder of RaiseWikibase, are quickly uploaded using the `page` function and can be easily adapted to a new use case.

The Wikidata properties are reused. Unfortunately, the federated properties in Wikibase are still under development and the extension WikibaseImport does not work as expected and turned out to be slow. We created three properties with IDs "P1"[3], "P2" and "P3" shown in Table 1a. Then, the Wikidata endpoint is queried for all (8600+) properties with labels, descriptions, aliases, datatypes,

---

[3] This is only the last and unique part of ID. The full URLs are omitted for brevity.

formatter URLs and formatter URIs for RDF resources. Those properties containing the triples with "P1"-"P3" are then created locally, see Table 1b.

| (a) Four properties created manually | | |
|---|---|---|
| BERD ID | Wikidata ID | English label |
| P1 | - | Wikidata ID |
| P2 | P1630 | formatter URL |
| P3 | P1921 | formatter URI for RDF resource |
| P4 | - | native company number |
| (b) More than 8600 properties queried from the Wikidata endpoint | | |
| P5 | P6 | head of government |
| ... | ... | ... |
| P8656 | P9448 | introduced on |
| (c) Automatically matched properties by semantic annotator "bbw" | | |
| P1020 | P1320 | Open Corporates ID |
| P91 | P159 | headquarters location |
| (d) Manually matched properties | | |
| P5699 | P6375 | street address |
| P588 | P813 | retrieved |

Table 1: (a) Four properties created manually, (b) more than 8600 properties queried from the Wikidata endpoint, (c) the properties matched automatically by the semantic annotator "bbw", and (d) the properties matched manually.

The German company dataset[4], donated to the Open Knowledge Foundation Deutschland by OpenCorporates, is converted to a CSV file. To automate ontology learning with Wikidata, the open source semantic annotator "bbw" [7] is applied to a part of the "company" table. The properties matched automatically are listed in Table 1c. Table 1d shows the properties matched manually. Additionally, we created a property for a native company number with ID "P4" as shown in Table 1a. It corresponds to the registration authority (court), the code related to a legal form and the number which is unique for the given court.

To reduce memory requirements, the JSON representations of entities are created using RaiseWikibase while reading the data from a CSV file line by line and the `batch` function is executed for lists with 100000 entities. Each entity has at most three claims with one qualifier and one reference. A million entities of German companies are filled into the BERD instance in seventy minutes.

Note that ElasticSearch indexing and building some of the secondary tables take additional time. We plan to add a multiprocessing implementation to improve on those issues.

---

[4] https://offeneregister.de

## 4   Conclusions

We presented the open source tool RaiseWikibase for speeding up data filling and knowledge graph construction using Wikibase and shared it at https://github.com/UB-Mannheim/RaiseWikibase. Up to a million entities and wikitexts per hour can be filled. A reusable example of knowledge graph construction is provided.

*Acknowledgments.* This work was funded by the Ministry of Science, Research and Arts of Baden-Württemberg through the project "Business and Economics Research Data Center Baden-Württemberg". We thank Jesper Zedlitz for [11].

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
