# OpenReview forum: "RaiseWikibase: Fast inserts into the BERD instance"
_eswc-conferences.org/ESWC/2021/Conference/Poster_and_Demo_Track — ESWC2021 P&D_

### Official Review · AnonReviewer1 · 2021-04-07
**Useful tool for WikiBase bulk load, minor corrections required.**

**Rating:** 7
**Confidence:** 4

**Review:**

The authors present a novel, open-sourced tool to load data into a WikiBase instance by directly inserting it to a database using transactions, thus reducing the required time. They also present a KB created with the tool.

The tool indeed seems to be very useful for bulk inserting to a WikiBase instance. It is listed in the MediaWiki article about WikiBase, it provides sufficient details of how to use the tool on github, and has been starred and forked, which indicates people are interested.

The paper requires a few clarifications and corrections. In Fig. 1, it is never indicated what the shapes included mean. In the third paragraph of Section 3, the definitions of properties could be moved to a table for quick reference and also improve the readability. Minor corrections: use \paragraph{} instead of \textbf{} in the introduction; it seems to be an extra space in the third paragraph of Section 2, when enumerating the functions, it is probably inside a \texttt{}.

**Anonymity:**

Yes, I would like my review to remain anonymous.

---

### Official Review · AnonReviewer4 · 2021-04-08
**Interesting engineering work but no scientnfic contribution**

**Rating:** 4
**Confidence:** 5

**Review:**

This short paper describes the RaiseWikibase tool, a software designed to import large data and build a knowledge graph in Wikibase. It is a technical description that addresses a very specific problem.

Per se, this is an interesting work that could be useful for other people. However, beyond a software engineering work, there is no actual scientific contribution, hence I believe it is not appropriate to the regular posters & demo track or a research conference. I think it would be more relevant in a workshop.

Misc.: the references need to be improved, e.g.: 3 has no journal/editor; 6, 7 and 8 are not in the regular ISWC tracks but in satellite events.



**Anonymity:**

Yes, I would like my review to remain anonymous.

---

### Official Review · AnonReviewer3 · 2021-04-14
**A performant solution to upload and construct a knowlege graph using Wikibase**

**Rating:** 7
**Confidence:** 3

**Review:**

This work presents a tool to insert data in a database and construct a Knowledge Graph by using the wikibase. The tool is available in the GitHub repository https://github.com/UBMannheim/RaiseWikibase.


While the motivation of the paper is well explained, the authors mention only one approach tackling the same problem, which is not clear how that differs from the implemented tool. Do they just provide a tool out of the related work mentioned in [11]. Therefore, I think that the contribution could be emphasized more.

# Strong points
1) Topicality: The demonstrated approach is relevant for the Semantic Web community and fits well with the topics of the conference,
2) Potential significance: The proposed tool is very well described. I particularly enjoyed it and also the readme. The user is informed about all the steps of the approach. In addition, the performance of constructing a knowledge graph by inserting data directly into the database is provided.


# Weak points
1) Clarity: The approach is not sufficiently described. I acknowledge that the authors have a limited amount of pages in this type of submission, but it is still possible to present more concrete details about the approach that is being demonstrated. For instance, I found more details and explanations in the GitHub repository about the “Wikibase Data Model and RaiseWikibase functions” than in the paper



**Anonymity:**

Yes, I would like my review to remain anonymous.

---

### Decision · Program_Chairs · 2021-04-19

Accept